# Tissue Engineering for Penile Reconstruction

**DOI:** 10.3390/bioengineering11030230

**Published:** 2024-02-28

**Authors:** Elissa Elia, Christophe Caneparo, Catherine McMartin, Stéphane Chabaud, Stéphane Bolduc

**Affiliations:** 1Centre de Recherche en Organogénèse Expérimentale/LOEX, Regenerative Medicine Division, CHU de Québec-Université Laval Research Center, Québec, QC G1J 1Z4, Canada; elissa.elia@crchudequebec.ulaval.ca (E.E.); christophe73.caneparo@gmail.com (C.C.); stephane.chabaud@crchudequebec.ulaval.ca (S.C.); 2Division of Urology, Department of Surgery, CHU de Québec-Université Laval, Québec, QC G1V 4G2, Canada; catherine.mcmartin.1@ulaval.ca

**Keywords:** tissue engineering, urology, penis

## Abstract

The penis is a complex organ with a development cycle from the fetal stage to puberty. In addition, it may suffer from either congenital or acquired anomalies. Penile surgical reconstruction has been the center of interest for many researchers but is still challenging due to the complexity of its anatomy and functionality. In this review, penile anatomy, pathologies, and current treatments are described, including surgical techniques and tissue engineering approaches. The self-assembly technique currently applied is emphasized since it is considered promising for an adequate tissue-engineered penile reconstructed substitute.

## 1. Introduction

The penis can be subject to various congenital conditions, e.g., hypospadias or acquired pathologies, e.g., narrowing. Medical or surgical treatments are currently available but should be improved to improve patients’ quality of care and life. In the most challenging cases, a new technology, tissue engineering, could make a difference. It could be beneficial to circumvent the lack of adequate tissue to graft and reduce potential comorbidities associated with the donor site. Much progress has been made since the appearance of this new technique, but much remains to be done. Currently, technologies have been developed to meet the clinical needs corresponding to the different tissues making up the penis and could ultimately lead to a complete reconstruction of this organ.

Penile development is a complex process starting during the fetal stage and pursuing until puberty. This process involves many biological components, such as enzymes and cytokines, which play an essential role in tissue development and maturation. Genetic and environmental factors can affect normal penile development, leading to specific anomalies and lack of functionality [1]. While some of these anomalies may only be aesthetic, some patients experience reduced or suppressed organ functionality. The penile malfunctions range from failed voiding to sexual and impaired fertility issues, which can also impact the psychology and sociability of patients [2]. The affected children typically require surgical treatment, which remains a tremendous aesthetic, functional, and anatomic challenge.

Furthermore, surgical techniques continuously evolve over the years, mainly due to the high rate of complications and recurrences [3]. In addition, each surgery requires a significant amount of adequate tissue, which could become problematic in cases of recurrence since the same donor area can only be harvested once. Therefore, due to the potential lack of available tissues to perform the surgeries, efforts have been made to think outside the box [4]. Whether cellular or acellular, many biomaterials have been tested to perform surgeries, but the resulting recurrence rate still needs to be lowered [4].

The other common urethral disorder is its narrowing, called stenosis or stricture when it affects the anterior urethra. Male urethral stenosis most often results from injury, instruments, non-infectious inflammatory conditions of the urethra, hypospadias surgery, and, finally, sexually transmitted diseases [5]. Urethral strictures result in more than 5000 inpatient visits yearly in the USA. Yearly office visits for urethral stricture numbered almost 5 million between 1992 and 2000 [6]. The total cost of urethral stenosis-related diseases in the United States was approximately $300 million in 2010. The annual cost of health spending is still increasing by more than 6000 USD per person after a diagnosis of urethral stenosis [7]. Patients with urethral stricture also appear to have a higher rate of urinary tract infection (41%) and incontinence (11%) [8].

In this review, we describe the penile anatomy, the different common penile pathologies and anomalies, and the current surgical options. On the other hand, the ascending impact of tissue engineering and its various applications will be discussed. However, any cell therapy treatment and cosmetic surgeries will be excluded from this article.

## 2. Penile Anatomy

An overview of the normal penile anatomy will be given in this paragraph. In addition, vascular and neural anatomy will be briefly discussed due to their critical roles in penile function. Indeed, this anatomy must be considered during penile reconstruction. Otherwise, dysfunction and recurrence may result in some patients.

The human penis is the male organ of urination and copulation and is composed of an anterior and a posterior urethra (Figure 1A). It includes different tissues whose roles are critical to maintaining their functionality. Two dorsolateral corpora cavernosa are located side by side on the dorsal part of the penis (Figure 1B). Cylindrical in shape, they are part of the erectile tissues. They become engorged with blood during the erection phase, allowing the penis to stand up and maintain itself. The tunica albuginea is an extensible tissue rich in elastin fibers, which envelope the corpora cavernosa. By constricting the deep dorsal vein, blood cannot leave the penis, and the erection is maintained.

The corpus spongiosum is located on the ventral side of the corpus cavernosum and is part of the erectile tissues. It surrounds the urethra from the prostate to the urethral meatus, which allows the urine and semen to be transported. The male urethra is divided into four regions, internally lined by different waterproof epithelia that avoid contact of urine with the neighboring tissues. The prostatic urethra is lined with transitional epithelium (urothelium). The membranous urethra is a transition region between the urothelium and pseudostratified columnar epithelium. The bulbous urethra and the pendulous urethra are lined with pseudostratified columnar epithelium. The fossa navicularis, the distal portion of the urethra, just proximal to the external urethral orifice (meatus), is lined with squamous epithelium continuous with that of the glans [9]. This cellular organization is also found in the oral mucosa. These cells lay on the basal lamina and lamina propria, which allows innervation, vascularization, and exchanges between the epithelial cells and the underlying tissues [10]. A muscular layer surrounds the urethra, allowing it to compensate for the increase in pressure during the passage of urine and facilitate its excretion.

The penile vasculature comprises three arteries: cavernosal, dorsal, and bulbourethral. They all arise from the internal iliac artery through the internal pudendal artery. The deep dorsal artery enables glans spongiosum enlargement during erection. The cavernosal artery allows corporal enlargement. Those three arteries communicate as a network through an anastomosis near the glans. Only one deep dorsal vein above the tunica albuginea provides the blood a way out. Blood exit is possible through drainage from the corpora cavernosa with emissary veins and from the corpus spongiosum with circumflex veins [11].

Penile innervation is composed of dorsal, cavernosal, and perineal nerves. Dorsal nerves transport sensation to the penile skin. The perineal nerves supply the ventral shaft skin, the frenulum, and the bulbospongiosus muscle. Cavernosal nerves are believed to exchange signal communication implicated in erectile function [12]. Due to this anatomic specificity, correction of congenital/acquired penile malformation like curvature or stenosis can result in nerve damage and dysfunction.

In the next part, the related common penile pathologies will be discussed.

## 3. Common Penile Pathologies, Anomalies, and Treatments

Pathologies affecting the urethra can be classified into two major groups: congenital or acquired [13] (Figure 2). As previously discussed, penile development is a complex process starting during the fetal stage and pursuing until puberty with many involved critical factors. In addition, a significant proportion of genitourinary malformations affect the penis. Congenital and acquired anomalies of the penis usually involve anatomic abnormalities and aesthetic discomfort. In addition, surgical repair is needed when the function is impaired or cosmetic correction is desired. Several excellent reviews have been written and can be read with profit [14,15,16,17,18,19,20].

Urethral reconstruction has received much attention recently due to the recurrence of urethral strictures after treatments. Various surgical techniques have been developed [41]. Urethral stricture can be treated by urethral dilation and urethrotomy. However, in long urethral strictures, substituting urethroplasty with genital skin and buccal mucosa grafts is the only option [42]. The current gold-standard therapeutic approach for complex urethral strictures usually involves reconstruction with autologous tissue from the oral mucosa [43]. However, surgical procedures are associated with severe complications, such as fistulas, urethral strictures, and dehiscence of the repair or recurrence of chordee [44,45]. Moreover, there are multiple situations whereby buccal mucosa is inadequate or inappropriate for utilization [46].

Penile tissue fibrosis can be treated using cell therapy: mesenchymal stem cells (MSCs) have proven to be the most effective, with a plethora of preclinical data suggesting the benefit of stem cells for use in penile fibrosis [47]. Stem cell therapy has a promising future in pediatric urological conditions [48,49].

Priapism treatment often uses an intracavernosal injection of a sympathomimetic agent and shunt procedures that create a connection with the corpus cavernosa and a neighboring structure [50]. A novel penoscrotal decompression technique has recently been described. Early treatment with super-selective embolization can be required [51]. Surgical interventions have consisted primarily of penile revascularization surgery for arterial insufficiency and penile venous surgery for corporoveno-occlusive dysfunction, whatever the mechanism.

Penile vascular surgery for treating erectile dysfunction (ED) is still regarded cautiously, but the surgical effectiveness remains debatable and unproven [52,53,54]. Furthermore, no complete restoration of erectile function is possible with the currently available techniques [55].

As for Peyronie’s disease, allotransplantation and penile implants are emerging strategies for treating all kinds of penile anomalies [17,56]. However, even if patients demonstrated viable grafts and natural erection, some patients developed a urinary fistula and graft rejections [57], which correlated with the results of the first penile transplantation in the USA [58]. In addition to needing re-operations, penile transplantation requires lifelong immunosuppression, exposure to malignancies and opportunistic infections for a non-life-threatening disease [59].

Despite the use of a variety of tissues to correct penile pathologies, two severe limitations of the currently used treatments exist: availability of a sufficient amount of adequate tissues and post-graft complication. For example, we highlight the case of urethral repair. Markiewicz et al. reviewed 1267 studies on the use of oral mucosa in reconstructive surgery from 1966 to 2006. Oral mucosa-based urethroplasty has been used in 1353 cases for urethral strictures and hypospadias/epispadias [15], with a success rate of 66.5% and 76.4% [60]. However, many complications are encountered. The removal of this tissue has disadvantages in more than 80% of patients [61], such as pain, numbness, submucosal scars, dry mouth, lesions [62], neurosensitive defects, discomfort [63], disorders of the opening of the mouth (contracture), and risks of oral infections. In addition, compared to the autologous urethral tissue, these substitutes have a limited barrier function for urine, a particularly toxic product. This can then lead to complications (strictures, failure of the transplant, etc.) [64]. In addition, even if mucosa can be harvested from both cheeks, the amount of tissue that can be harvested from a donor site is limited, which can be problematic, especially in the case of prolonged urethral deficits. It should also be noted that the mucosa cannot be harvested twice from the same site, further limiting surgical options in the event of complications or recurrence of the pathology [65,66,67]. Donor site morbidity is 16 to 32% for buccal mucosa graft [68]. To overcome these difficulties, alternative methods for urethral reconstruction are required. Tissue engineering can be a solution. Indeed, using tissue-engineered grafts may prevent complications, thus reducing health costs and improving the patients’ quality of life.

## 4. Tissue Engineering

Tissue engineering is an expanding scientific field that involves many scientific disciplines, such as cell biology, chemistry, physics, toxicology, mechanical and electrical engineering, etc. Its main objective is reconstructing tissues or organs in vitro [69,70,71,72]. These substitutes can be used for clinical applications such as grafts or biological dressings. They can be used to produce models for studying physiological or pathological biological processes [73,74].

Its principle is based on the association of four elements (Figure 3): first, the cells can present more or less stemness and come from different stages of development [75,76]. Nevertheless, for clinical applications, autologous cells should be favored. The ideal for research models is to use cells from the animal and the target organ when available: horse lung cells to study equine asthma [77] or cells from the human bladder to model bladder cancers [78].

The second element is the scaffolding, which is used to give shape to the tissue/organ to be reconstructed but whose nature has significant implications; in particular, it provides adequate mechanical resistance or signals promoting proliferation and differentiation of the cells, which will be used [79,80,81,82]. There is a vast variety of materials that can be used and have been described [16,44].

The third element that enters the equation is the technique used to produce the scaffold [16]. This can significantly influence the physicochemical parameters of the resulting tissue, even with identical biomaterial. For example, the porosity of a scaffold produced by electrospinning or moulding can be substantially different, just as the cohesion of other materials can be improved depending on the technique used. Techniques can also be combined to produce a result closer to expectations.

The fourth and final element we often underestimate is the culture media in which the cells will be cultivated [83,84,85,86,87]. These media not only allow the cells to proliferate on the surface or inside the material by providing it with signals, often in the form of growth factors, but also allow the cells to differentiate to play their role, such as the formation of a barrier to urine for epithelial cells [88,89] or the formation of a muscular structure to allow contractions [90,91].

In this review, we will mainly discuss the materials used.

### 4.1. Urethroplasty

#### 4.1.1. Synthetic Biomaterials

Non-degradable biomaterials: Non-degradable biomaterials such as silicone and polytetrafluoroethylene (PTFE) have been tested for tissue reconstruction. Many complications have been found, such as calcification, fistula formation, chronic hematuria, formation of stones, and significant contraction up to 50% of the initial length. The success rate is meagre, explaining these biomaterials’ fall out of favor [92,93].

Degradable biomaterials: Research then turned to degradable biomaterials, allowing tissue growth while the scaffold resorbs over time in a controlled manner. Synthetic biomaterial biodegradation duration generally varies from several days to months [94]. Incorporating cells into the scaffold improves the biocompatibility and maintenance of the tissue because the cells contribute to the remodeling process. It is mainly due to the synthesis of the components of the extracellular matrix, which are essential for the long-term survival of the implanted construction. Indeed, proper synchronization between the biodegradation of the scaffold and the production of cellular components is crucial for the success of the transplant. In addition, the substitutes must match the replaced tissue’s mechanical characteristics and allow its manipulation and suturing [95].

The most studied synthetic biomaterials are linear polyesters such as poly-lactic acid (PLA), poly-glycolic acid (PGA), poly-lactic-co-glycolide (PLGA), and copolymers and polycaprolactone (PCL) [96]. They have the advantage of being biocompatible 3D scaffolds with low cost and malleable mechanical properties. Therefore, they allow rapid and reproducible results with a low risk of contaminants, available every time (some being off the shelf). In addition, they will enable the incorporation of substances such as growth factors [94]. Despite this, these biomaterials have drawbacks, mainly the long-term effects of matrix degradation products, which are little known and can, for example, exacerbate inflammatory reactions. In addition, the environment is inadequate for the differentiation and organization of epithelial cells, which can lead to non- or poorly functioning tissue, allowing the passage of urine through the tissue and, therefore, the formation of stenosis through inflammation and the graft [97,98].

#### 4.1.2. Intelligent Biomaterials

“Intelligent” biomaterials have recently emerged, reversibly responding to temperature, ionic strength, pH, or light [94,99,100]. Signaling molecules, such as extracellular matrix components or growth factors, can be loaded onto these biomaterials. Delivery can be triggered using external stimuli (e.g., pH, temperature, or light) or carried out as the scaffolds are biodegraded [101]. These intelligent biomaterials are mainly studied for the administration of drugs and medical device applications [94] but can be adapted for urological use. Thermosensitive polymers represent the most significant part of these biomaterials. Depending on the temperature, the solubility of thermosensitive polymers in aqueous solutions is modified. The most temperature-sensitive polymer in water is poly (N-isopropylacrylamide) (PNIPAM) with a critical temperature of the solution lower than 32 °C, which makes the polymer particularly applicable in biology [102]. Shape Memory Polymers are another intelligent biomaterial first studied in vascular and bone tissue engineering [103,104]. These materials were considered relevant in the context of urinary stimuli. Indeed, these polymers have the distinction of existing in an original form and being able to stay temporarily in another shape. The return to the original condition occurs upon exposure to a stimulus, usually heat [105]. In the urethral context, it can stretch during erection and return to its original form during detumescence, which gives it interest. Other intelligent biomaterials are still in the early stages, such as acellular heparin-collagen with growth factors [106] or the stress-induced rolling membrane [107].

#### 4.1.3. Natural Biomaterials

Silk fibroin: Other natural polymers have been tested for urethral reconstruction, particularly polymers derived from Bombys mori cocoons, such as silk fibroin [108,109], previously used to make sutures. Their main advantage compared to other natural and synthetic biomaterials is their excellent physical characteristics (elasticity and resistance to tearing) [94]. Panilaitis et al. have shown that, in comparison to SIS, silk fibroin induces fewer immunogenic and inflammatory responses, suggesting better biocompatibility compared to conventional urological biomaterials [110]. Chung et al. demonstrated that using a bilayered silk fibroin scaffold in rabbits allowed tissue regeneration like traditional SIS matrices with reduced immunogenicity. However, this biomaterial comprises only two elements: silk fibroin (75%) and sericin (25%), which remains far from the native tissue.

Furthermore, biodegradability is very long or even nonexistent [111]. Silk fibroin/keratin films improved mechanical properties when blended with gelatin. After mixing with calcium peroxide, an oxygen-generating substance, the films generated high oxygen levels, promoting enhanced cellular growth and demonstrating antibacterial ability. A preclinical trial showed the feasibility of applying this scaffold for repairing urinary tract defects. Overall, this study suggests that the keratin/silk fibroin scaffold has the potential to be a viable option for urinary tract tissue engineering, but further research is needed to fully assess its safety and efficacy in vivo [112]. Recently, a bi-layer silk fibroin graft for tubular urethroplasty in a porcine defect model was developed [113]; such a material could be tested for urethroplasty soon.

Collagen: Collagen is also used to create scaffolds for tissue reconstruction. A study completed in 2016 by Pinnagoda et al. showed increased burst pressure and weak suture retention values when tubular acellular collagen scaffolds were used for urethral regeneration in rabbit models. Collagen scaffolds showed good biocompatibility and gradual regeneration with time. No postoperative complications were observed in 60% of all animals. However, stenosis and fistulae formation were seen in 40% of the animals (20% for each of these complications) [114]. Several improvements have recently been performed using these hydrogels, but further studies remain necessary to evaluate the potential of collagen type I for urethral reconstruction [115]. Collagen type I scaffolds need to be cross-linked to slow down enzymatic degradation and enhance their mechanical properties, but this could cause the loss of their natural characteristics [116].

Decellularized matrices (SIS, BAM): On the other hand, so-called “natural” biomaterials have been studied in parallel with synthetic ones. They include animal matrices or human cadaveric organs, which have been decellularized enzymatically, physically, or chemically to obtain an acellular matrix whose mechanical properties and biochemical environment are identical to native tissues. The main acellular matrices studied in vivo for repairing penile tissues are the “Small Intestinal Submucosa” (SIS) and the “bladder acellular matrix” (BAM) [117]. The SIS is mainly formed from the submucosa of the small porcine intestine, where the mucous, serous, and muscular layers are mechanically removed from the inner and outer surfaces of the intestinal wall to leave a 0.1 mm collagen-rich membrane [118,119]. In vivo, animal experiments and pilot studies in humans have shown that this material can replace the urethra thanks to rapid cell ingrowth and significant angiogenesis, comparable to skin and mucosal grafts [120,121,122]. Furthermore, these biomaterials are entirely biodegradable in four to eight weeks, can stretch under force, and have a low tendency to tear [123]. Recently, the decellularized urethra has been evaluated in vitro and proved to have the potential to be used for tissue engineering applications [124].

However, there exists a balance between a real immune risk due to the presence of residues of biological elements in these tissues (DNA, prion) [125,126] and reduced immunogenicity due to the high similarity of the decellularized matrix compared to the native tissue [127]. Another limitation is the limited vascularization of these tissues, which can induce necrosis of the implanted biomaterial. Furthermore, one of the significant problems with acellular matrices, as shown by the experience in rabbits by Dorin et al., is that urothelial regeneration on the surface of the acellular graft is limited to 0.5 cm, which compromises success in more complex cases, such as long urethral stenosis [128]. In addition, this biomaterial does not allow mature tissue to fulfill its waterproof function. It is, therefore, necessary, after urethral replacement, to carry out a period of “urinary diversion” to allow the tissue to mature and become functional in preventing extravasation of urine and early local irritation and, therefore, the risk of secondary inflammation and scarring [129]. Other techniques implementing hydrophobic multilayer SIS to reduce urine leakage have also been implemented, but these systems still need to satisfy all the necessary constraints.

Despite the apparent benefits of SIS, the clinical results have been poor, with infections being the most significant limitation [122,130]. The donor’s age and the intestine area from which the SIS matrix is derived impact the regenerative potential, which has prevented this biomaterial from being considered ideal for urethral repair [131]. Bladder Acellular Matrix (BAM) is another example of a decellularized matrix that has successfully shown regeneration of the urethra in vivo in rabbits [132,133]. Despite the rigorous and efficient decellularization process, as for SIS, residual immunogenic components are found in BAM, such as DNA, which can induce functional failures or even inflammatory reactions and, therefore, rejection of transplants [134]. Another limitation of this type of biomaterial is the significant variability from one batch to another, necessitating process improvement before BAM can be applied in tissue engineering [94].

Despite the moderate success of degradable biomaterials in preclinical studies, the transition in the patient remains a significant challenge today. Natural and synthetic materials have been combined to produce hybrid biomaterials that better meet clinical performance requirements. The properties targeted for tissue engineering applications are mechanical resistance, cellular affinity to attract surrounding cells, porosity, biocompatibility, and biodegradability to renew the extracellular matrix. Current efforts are mainly focused on determining the appropriate biomaterial and its synthesis for applications in biology.

### 4.2. Tissue Engineering for Peyronie’s Disease Correction

To circumvent these inconveniences, natural and synthetic biomaterials have been tested. Despite the advantages of polytetrafluoroethylene (PTFE) and polyethylene terephthalate (PET) (cost, availability, low risk of contamination, etc.), severe fibrosis, inflammation, impotence, and low satisfaction rate have been found and led to loss of interest in them [135]. Ferretti et al. compared an acellular or cellularized PGA scaffold with autologous fibroblast in rats for Peyronie’s disease. They showed that, at four months, a significantly higher retraction was found in the acellular biomaterial than in the cellularized one. Furthermore, erectile response to cavernous nerve stimulation was significantly higher in the cellularized biomaterial [136]. Kershen et al. seeded smooth muscle and endothelial cells onto biodegradable polymers and successfully implanted them into athymic mice [137]. Despite progress, functional restoration of the corpora cavernosa with synthetic biomaterials has been limited.

Recently, a bi-layer silk fibroin graft has been used on a rabbit model as a substitute for tunica albuginea in a corporoplasty experiment. The results seem satisfactory, with a reduction of the expression of contractile proteins with a lower extent of fibrotic tissues, contrary to what is observed when SIS or tunica vaginalis flaps were used [138].

Concerning the natural biomaterials, Knoll et al. tested SIS as a graft to correct penile curvature on 148 patients with a follow-up of 6 to 96 months. Curvature recurrence appeared in only 9% of patients, with a postoperative complication of only 5%. More than 87% of patients indicated having no erectile dysfunction [139]. However, other studies have shown a much higher complication rate of 37%, including infection, curvature recurrence, and hematoma [140]. The cadaveric pericardium has been tested in 11 patients with a resolution of curvature in 81%, but long-term follow-up resulted in difficulties in maintaining an erection [141]. Another study using pericardial graft in 81 men resulted in 75% satisfaction with the same or better postoperative penile rigidity in 68% of patients [142]. Acceptable results have been obtained using cadaveric dura mater graft, but the report indicated the risk of contracting Creutzfeld–Jacob Disease [143].

Cadaveric and xenograft materials are commercially available for Peyronie’s disease. However, poor structural integrity outcomes of tunica albuginea were found, which added fibrosis, inflammation, and contraction of the grafted area [144]. Allotransplantation [145] and penile implants [146] are emerging concepts which are highly effective in restoring penile functionalities, but life-long immunosuppression and considerable costs are the main disadvantages. Finally, the application of commercially available hemostatic agents like high-density collagen gel tubes has been tested in a rabbit model without suturing [147]. However, residual curvature has been found with weaker erections and resultant fibrosis.

### 4.3. Corpus Cavernosum Replacement Strategies

Excellent reviews have been published on this subject and can be referred to for more information [17,148]. Many studies have been undertaken to reconstruct an adequate substitute for the corpus cavernosum. Initial studies were initiated by Atala’s group, who seeded human corporal endothelial cells (EC) and smooth muscle cells (SMC) on biodegradable polyglycolic acid (PGA) scaffolds characterized by high tensile strength, high melting point, and degradation properties. Although these scaffolds contributed to a well-vascularized SMC and EC growth after implantation into mice, they could not mimic the three-dimensional penile structures. Acellular matrices (ACM), similar to native tissues, were prepared by decellularizing the extracellular matrix to improve the structure. Subsequently, the organization of SMC and endothelial cells increased, forming the vascularized corpus cavernosum. However, the vascularization pattern differed from native human tissue as this engineered corpus cavernosum was formed ex vivo before implantation. This might harm cellular density and erectile function [17]. Chenyang Ji et al. demonstrated that muscle-derived stem cells (MDSCs) seeded on acellular corporal collagen matrices (ACCMs) could reconstitute a functional corpus tissue in vivo. However, different degrees of fibrosis and necrosis were noticed due to a lack of capillaries and adequate nutrition for the MDSCs. More advanced studies focused on promoting the formation of blood vessels to improve MDSCs’ resistance to hypoxia. A lentiviral vector transducing method was elaborated to create MDSCs expressing vascular endothelial growth factor (VEGF). These cells showed better attachment and growth than untransfected cells, which indicates that VEGF plays a vital role in developing a vascularized corpus cavernosum tissue in vivo and promoting the development of MDSCs [149].

Another study by Geng An et al. was based on using three-dimensional (3D)-printed hydrogel scaffolds with porous structures similar to those of penile cavernous tissue, providing a suitable microenvironment for cell adhesion, growth, and migration. Heparin, a polysaccharide with a good affinity for angiogenic factors such as VEGF, was added to the 3D-printed hydrogel scaffolds. On the other hand, hypoxia-inducible factor-1α (HIF- 1α)-overexpressed MDSCs were created by lentiviral transfection. HIF-1α-mutated MDSCs promoted neovascularization since HIF-1α is an essential transcription activator for regulating vascular growth factors. Heparin-coated scaffolds seeded with HIF-1α-mutated MDSCs were implanted into injured corpora cavernosa. As a result, angiogenesis induced by HIF-1α and VEGF repaired the injured corpora cavernosa and restored penile erection and ejaculation function in vivo [150].

### 4.4. Bioprinting

Three-dimensional bioprinting is a tissue reconstruction technique where the cells are integrated into a cross-linkable hydrogel matrix called bioink to create 3D tissue equivalent constructs in the desired pattern. Three-dimensional bioprinting requires essential components such as 3D imaging, bioink, and a bioprinter [151].

The leading technologies used for deposition and patterning biological materials are inkjet, microextrusion, and laser-assisted printing.

Inkjet bioprinting: This technique involves the deposition of tiny droplets of bioink, a mixture of living cells and a biocompatible material, such as a hydrogel, using inkjet printheads. The droplets are ejected from the printhead and deposited onto a substrate in a precise pattern to create a 3D structure. This method is relatively inexpensive and can produce high-resolution structures, but the high shear forces during droplet ejection can damage the cells [152].

Microextrusion bioprinting: This technique uses a syringe or a pneumatic device to extrude bioink through a nozzle onto a substrate, layer by layer. This technique can print various cell types and materials, including viscous and high-density materials. However, the resolution is limited, and the shear stress during the extrusion process can damage the cells [152].

Laser-assisted bioprinting: This technique uses a laser to create a pressure wave that ejects a droplet of bioink onto a substrate. This method allows for the precise deposition of tiny droplets of bioink and can create complex structures with high spatial resolution. However, the high cost of the equipment and the limited availability of suitable bioinks are some of the challenges associated with this technique [152].

This technique remains challenging despite the benefits of 3D tissues with neovascularization ability. The main challenges include culturing the printed construct with a different supporting medium in a bioreactor or with any particular maturation setup. On the other hand, nowadays, the highest possible printing resolution with a laser bioprinter is 20 µm, while small capillaries of native tissues are 3 μm in diameter. In addition, the complexity of the vascular networks has not yet been achieved in 3D bioprinting. Other challenges are the requirement of a high number of cells, extensive research on optimal bioink composition with good printing abilities, the shape fidelity, and the demand for sophisticated advanced complementary bioprinting technologies with ensured environment and sterility maintenance. Bioprinting has several other difficulties, such as safety, affordability, large-scale production for clinical translation, and ethical issues that include the cell source and its processing procedures [151]. Bioprinting is not currently used for penile reconstruction, but its application may be promising.

### 4.5. Acellularization of the Penis

Reconstruction for total penile defects is considered a unique challenge due to the anatomic complexity of the penis. Decellularization helps maintain the structure and functionality of the native organ and the extracellular matrix (ECM) architecture and limits immunogenic responses. In addition, decellularized organ scaffolds provide biomechanical support for cell adhesion, proliferation, and differentiation. Yu Tan et al. developed the first protocol for decellularizing whole-organ human penile specimens for total penile tissue engineering. The scaffold reported in Yu Tan et al.’s study represents the anatomical complexity with an increased scaffold diameter due to the use of whole human organ specimens with corpora and urethra rather than a single subunit. To overcome this challenge, a hybrid decellularization scheme was applied, combining micro-arterial perfusion, urethral tube perfusion, and standard external diffusion, followed by the recellularization of the scaffold with adipose-derived stem/stromal cells (ASCs). This scaffold is characterized by a good ability to support cell attachment and growth. However, large-scale reseeding of the entire scaffold may present a significant challenge. Furthermore, multiple cell lineages and differentiation may be required to restore the complex cellular morphology of the penis. Decellularizing the entire human penis is the initial step toward developing a successful tissue-engineered human penile scaffold [153].

## 5. Tissue Engineering: The Self-Assembly Approach

In the last decades, a new tissue production method has been developed in LOEX, Canada: the “self-assembly” [154]. This technique is based on the ability of fibroblast-like cells to secrete and deposit their ECM to form cellularized sheets. The secretion of collagen by cultured fibroblasts in a medium supplemented with ascorbate was initially demonstrated by Sumner and Switzer [155], and the potential to form stroma by Senoo and Hata [156] a few years later. These two studies are the basis for producing reconstructed tissue with a scaffold free of any exogenous biomaterial. This technique has been successfully applied to produce organs such as blood vessels and skin for severely burned patients in the clinic [157]. Subsequently, other tissues, such as the cornea, were successfully reconstructed in clinical trials [158]. Concerning the applications in the urology domain, the self-assembly method allows the production of different substitutes, such as urethral substitutes, which can be used for penile reconstruction. However, many other substitutes can be produced, such as vaginal substitutes [159] and cancer models [160], but they will not be described here.

### 5.1. Urethral Substitutes

Based on the skin and blood vessel models reconstructed at LOEX, the use of the self-assembly technique has also been validated for the production of flat (Figure 4A) [161] and tubular tissues (Figure 4B) [162].

The formation of a urothelium with adequate differentiation and a water-tightness similar to the native tissue was then obtained by maturation either at the air/liquid interface or by conditioning in a bioreactor [162].

Tissue was stimulated or not in a bioreactor. However, whatever the condition, all the tissues, perfused or not, epithelialized or not, had sufficient mechanical strength for clinical application (Figure 5) (Table 1) [163].

These tissues have been implanted subcutaneously in mice and have shown excellent transplant and survival rates [164]. Endothelialization facilitated early inosculation and reperfusion, but its clinical benefit could not be demonstrated, probably due to the limited thickness of the tissues. Subsequently, transplants were performed in rabbits. Despite some technical challenges, these have been successful (Figure 6).

However, two significant unfavorable in vitro results were noted: stiffness or lack of elasticity and keratin 14 (K14) expression in the epithelial layer, which should be absent from a mature urothelium but restricted to progenitor cells [165]. The native empty urethra is arranged as a star, the branches expanding as the pressure generated by the urine flow increases during urination [166]. This allows maximum pressure to be absorbed. An overly rigid urethral tube could cause functional obstruction with long-term harmful effects on the bladder. Therefore, they focused on using mesenchymal cells extracted from the urinary tract to rebuild the tissues [165]. However, this is a challenge because these mesenchymal cells deposit significantly less extracellular matrix than dermal fibroblasts. A recent study by Caneparo et al. showed the variation in the strength, the strain failure, and the elastic modulus of the substitutes reconstructed with different percentages of dermal (DF) and vesical fibroblasts (VF) [167]. The strength, as well as the strain failure, increased with the rate of dermal fibroblasts included in the mix. At the same time, the elasticity, which is the opposite of the elastic modulus, decreased. These results helped to identify the best cell composition. Combining 80% VF and 20% DF gives the optimal tissue strength, resistance, and elasticity (Figure 7) [167].

The first set of experiments used DF to produce the stroma. In this context, K14, a protein usually expressed only in specific cells of the basal layer, was expressed throughout the urothelium. The expression of K14 could be attributed to the use of DF, which are non-specific cells of the urinary organs, leading to non-specific signals for urothelial cells. Studies at LOEX have shown the importance of reconstructing tissue with mesenchymal cells from the target organ to obtain adequate differentiation under physiological (cornea or bladder) [165,168] or pathological (psoriatic skin) conditions [169]. Bouhout et al. showed that contrarily to the DF-derived matrix, which led to a urothelium with an abnormal structure, the VF-derived matrix allowed the development of a urothelium that shared the phenotype of the native tissue [165]. A hybrid tissue mixing both cell types showed a limited level of urothelial maturation, intermediate between both previously observed phenotypes. These observations indicated that VF-derived stroma supports the urothelial development, while the DF and the 1:1 mix population matrices led to hyperproliferation of the urothelial cells rather than their differentiation (Table 2).

Indeed, using urological cells has led to more elastic tissues and a well-differentiated urothelium without the over-expression of K14 [165].

### 5.2. Tunic Albuginea Substitutes

As mentioned before, skin substitutes produced by the self-assembly technique are already successfully applied to severely burned patients in the clinic [157]. In urology, corpora cavernosa must correct the volume deficit to ensure the penile length and, therefore, allow voiding in the standing position, permit sexual activity, and finally have a satisfactory cosmetic appearance. To fulfill the demand for material for penile reconstruction in the case of Peyronie’s disease, autologous tunica albuginea has been produced and tested using the self-assembly method [170]. Imbeault et al. showed that the tissue-engineered endothelialized tubular graft was mechanically sufficiently resistant for clinical use. Furthermore, its structure was similar to the normal tunica albuginea. However, further functional testing remains necessary.

## 6. Limitations and Perspectives

Because of a lack of penile tissue for surgeries, many tissue engineering strategies have been set up (Table 3), with their advantages and inconveniences. Due to the high morbidity rate in patients and the lack of durable long-term results, an autologous urethra produced in vitro from a small biopsy may be the solution. Autologous cells would provide better growth potential, vital for correcting congenital anomalies. The self-assembly technique offers a living autologous tissue free of exogenous materials and their inconveniences. Since the cells are autologous, fewer adverse events may be expected following the graft. However, the preparation time of this tissue shall be taken into consideration. The reconstruction process is about three months, from the initial biopsy to cell culture and the completely mature and functional tissue for the graft.

Given that most patients must wait months before being operated on and that these operations are not threatening the patient’s life, the delay does not represent a significant inconvenience. To overcome the need to collect bladder cells from patients, induced pluripotent stem cells (iPSCs) may, in the future, be differentiated from patient blood samples into different cell types to reconstruct the desired tissue. Thus, the invasiveness of the operation could be replaced by a simple blood sample. However, tumor and mis-differentiation must be monitored at all differentiation steps [171]. Production costs must also be considered because autologous tissue engineering production is much more expensive than a simple acellular biomaterial. However, fewer complications are expected with greater patient acceptability and less cost after implantation. Paying more to produce this tissue should save some of the additional surgeries.
bioengineering-11-00230-t003_Table 3Table 3The different types of tissues, materials, and methods applied for penile reconstruction.TypeYearAnimalOutcomeReference**Tissues**



Oral (buccal, lingual) mucosa189019921992HumanDog/HumanHuman+[172][173][174]Genital skin19532008HumanHuman−[175][176]Dermal grafts19791995HumanHuman−[177][178]Vein graft1998Human+[179]Tunica vaginalis1995Human−[180]Fascia2022Human+[181]**Materials**



 *Synthetic* *20122004In vitroIn vitro/mouse+/−[97][98]Intelligent biomaterials:20132015In vitroIn vitro+[102][103] *Natural* Acellular matrices -silk fibroin-collagen2007200720132012201520142018HumanHumanHumanRabbitRabbitRabbit+/−[120][121][122][132][133][111][115]**Penile Decellularization**2019In vitro+[145]**Self-assembly technique**20112013MouseMouse+[170][164]In the outcome column, − is for negative outcome and + for positive outcome; −/+ are for outcome presenting positive and negative aspects. Other tissue engineering techniques have been used to reconstruct other organs and may be applied for penile reconstruction in the future, such as cell sheet production. The outcome in the table represents an overview of the results of the different cited articles related to the same type of tissue, material, or technique. The list presented above represents the trend in penile tissue engineering in the last decades. For a more exhaustive list, see references [17,42,44,49,94,148,182]. * PLA, PGA, PLGA, PCL.


## 7. Conclusions

The penis is a complex organ whose reconstruction requires different techniques. These techniques exist, but work has yet to be done so far to respond to the complexity of the organ as a whole. Among the reconstruction strategies currently applied, self-assembly is one of the most interesting.

In conclusion, patients’ quality of life presenting congenital or acquired penile anomalies should be significantly improved with an autologous tissue free of exogenous biomaterials reconstructed by the self-assembly technique. Despite the few inconveniences of this technique, the tissue quality produced should allow patients to return to a potentially more normal life.

## Figures and Tables

**Figure 1 bioengineering-11-00230-f001:**
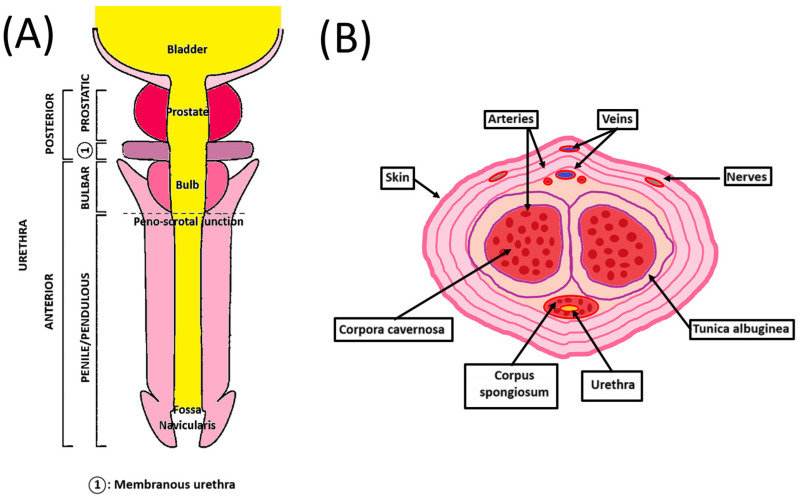
**Schematic penile anatomy.** (**A**) Penile top view showing the different parts of the anterior and posterior urethra. (**B**) Penile cross-section view showing the organization of the different components of the penis.

**Figure 2 bioengineering-11-00230-f002:**
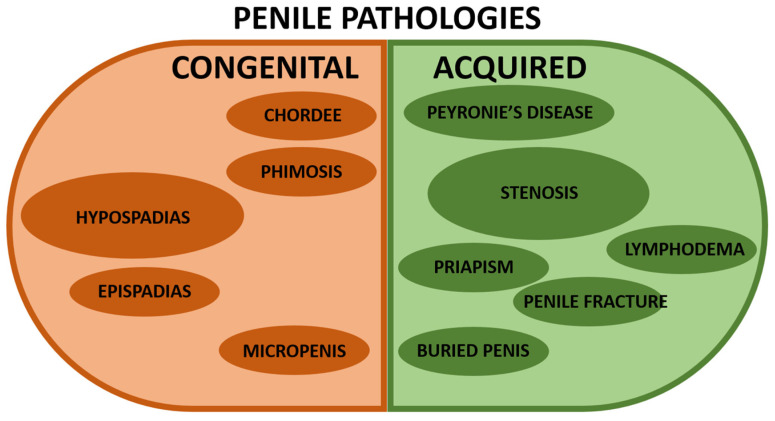
**Representation of the most common penile pathologies.** The left/brown half is for congenital penile anomalies, whereas the right/green half is for acquired penile pathologies. A larger size of the circle indicates a more significant incidence in the population (without being proportional). The references describing the pathologies are shown below. Congenital penile pathologies: hypospadias [1,20,21,22,23,24,25], chordee [26], epispadias [27], phimosis [28,29,30], and micropenis [31,32]. Acquired penile pathologies: Peyronie’s disease [33,34], stenosis [6,8], lymphoedema [35,36,37], priapism [36,37,38,39,40], penile fracture [37], and buried penis [30].

**Figure 3 bioengineering-11-00230-f003:**
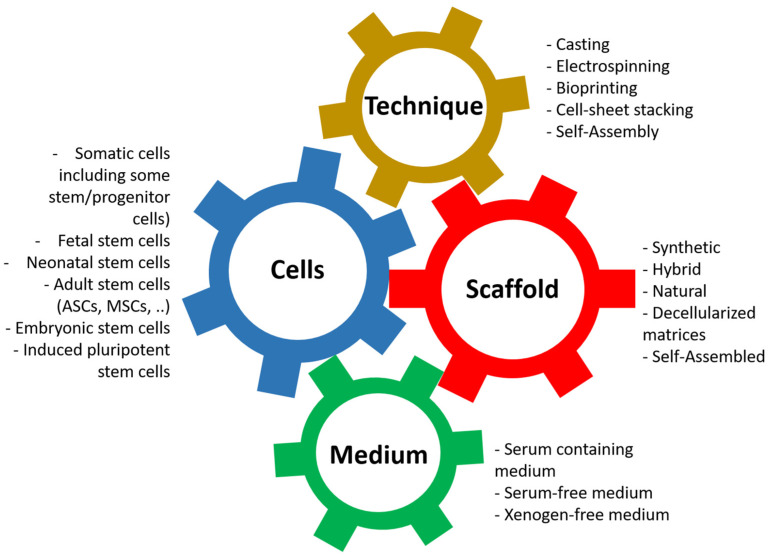
**Principles of tissue engineering.** Tissue engineering is the association between the cells, the scaffold, the technique, and the cell culture medium.

**Figure 4 bioengineering-11-00230-f004:**
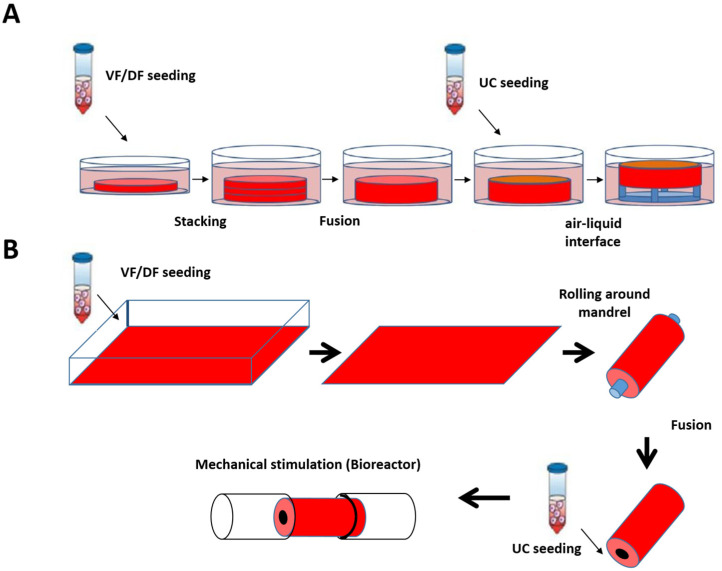
**Schematic protocol of urethral substitute production by the self-assembly method (flat and tubular models).** (**A**) Preparation of flat tissue (patch): The stromal cells are seeded on 6-well plates containing a paper anchor used to limit contraction and to facilitate tissue manipulation. A support stabilizes this anchoring. The cells are cultured in a medium supplemented with 50 μg/mL of ascorbate. On day 28, the sheets are detached from the plastic, and three sheets are superimposed to form a thicker tissue subjected to limited compression to allow the fusion of the cell sheets in 4 days. Then, the urothelial cells are seeded on the upper part of the construct and cultivated for seven days in submerged condition. The constructions are then raised at the air/liquid interface using a support for an additional 21 days to allow the epithelial cells to mature. (**B**) Diagram of the “tubular” model production by the self-assembly technique. Classic tubular self-assembly technique. The dermal fibroblasts are seeded in a cell culture plate covered with gelatin. They are grown for 28 days in the presence of ascorbate. The stromal sheet formed is tightly wound around a cylindrical mandrel. The resulting construction is cultivated for four days to ensure the fusion of the sheets. Once the fusion is complete, the mandrel can be removed, and the tubular structure can be perfused to be seeded with urothelial cells. The tube is matured in a bioreactor.

**Figure 5 bioengineering-11-00230-f005:**
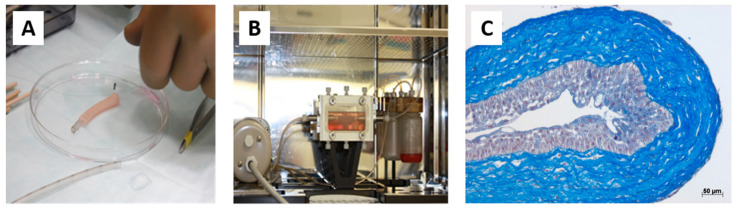
**Production and characterization of a tubular urethra produced in the laboratory by tissue engineering with dermal fibroblasts.** (**A**) The macroscopic appearance of the tubular tissue was reconstructed after rolling a stromal sheet around a mandrel. (**B**) After urothelial seeding, the tubular tissue is cannulated and put into a bioreactor for maturation. (**C**) Histological characterization of the tubular tissue after maturation (Masson’s trichrome staining): a well-differentiated urothelium is visible with basal intermediate umbrella cells.

**Figure 6 bioengineering-11-00230-f006:**
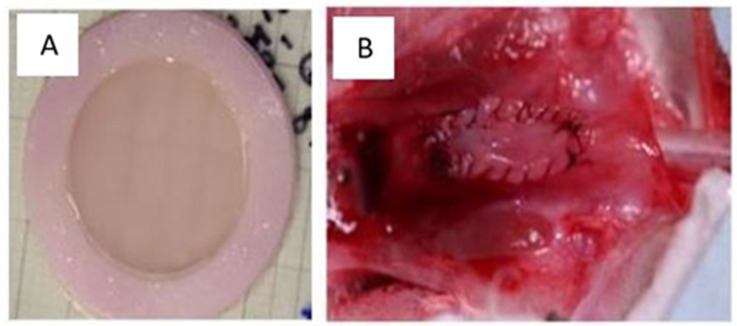
**Macroscopic appearance of the tissues produced.** (**A**) The appearance of flat human urothelial tissue after production in the laboratory. (**B**) A flat urothelial substitute implanted on the rabbit urethra.

**Figure 7 bioengineering-11-00230-f007:**
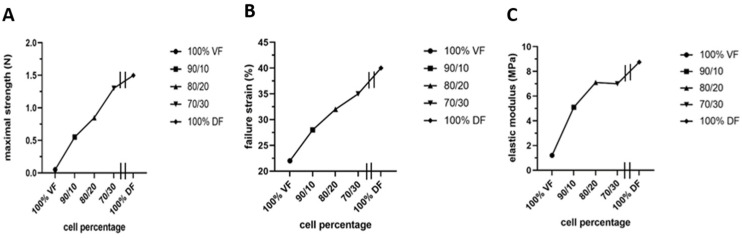
**Characterization of the mechanical properties of the substitutes produced with different ratios of vesical or dermal fibroblasts.** Each symbol represents the mean of measurements of one substitute. The 100% VF refers to the condition where the stroma of the urethra substitute was reconstructed using vesical fibroblasts (VF) only. The 90/10 is for a mix of 90% VF and 10% dermal fibroblasts (DF), 80/20 is for a mix of 80% VF and 20% DF, 70/30 is for a mix of 70% VF and 30% DF, and 100% DF is for the condition where the stroma of the urethral substitute was reconstructed using only DF. All the substitutes were produced using the hybrid technique of assemblage. (**A**) Maximal strength was measured using Instron ElectroPuls E1000 (in N). (**B**) Failure strain in (%). (**C**) Elastic modulus (inverse of elasticity) (in MPa).

**Table 1 bioengineering-11-00230-t001:** Burst pressure of the tissue-engineered tubular genitourinary grafts compared to the native porcine urethra.

Perfusion	− − −			+	+	Native porcine urethra
Urothelial cells	− − −			−	+	
Weeks	2	3	4	3 + 1	3 + 1	
Burst pressure (mmHg)	803	1133	1801	1761	1703	418

Perfusion: − indicates no perfusion into bioreactor was performed; + indicates perfusion into bioreactor was performed for the indicated time period in weeks. Urothelial cells: − indicates when no urothelial cells were seeded into the tubular constructs; + indicates when urothelial cells were seeded into the tubular constructs.

**Table 2 bioengineering-11-00230-t002:** Evaluation of the urothelium maturation on stroma of different origins.

	Native Urothelial Tissue	Reconstructed Urothelial Tissue (VF-Matrix)	Reconstructed Urothelial Tissue (VF:DF-Matrix)	Reconstructed Urothelial Tissue (DF-Matrix)
Presence of cells	+++	+++	+++	+
Differentiation of cells	+++	+++	+(+)	−

(+++): presence and/or differentiation of the three urothelial cell layers; (+): presence of one urothelial cell layer; (−): absence of differentiated cell layers.

## Data Availability

Not applicable.

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
