# Peer review of "Tissue Engineering for Penile Reconstruction"

_bioengineering, 2024, doi:10.3390/bioengineering11030230_

Round 1

Reviewer 1 Report

Comments and Suggestions for Authors

This paper is a comprehensive review on tissue engineering in penile reconstructive surgery. The authors present an overview on congenital and acquired penile anomalies and their treatment, emphasizing the role of various tissue engineering and future perspectives. They have included most relevant studies in the field and designed respectable analysis. A special attention has been made on self-assembly technique, and I believe this would make an additional benefit for the readers. Finally, I would congratulate authors for this well-designed, comprehensive paper on a very interesting and up-to-date topic. 

Reviewer 2 Report

Comments and Suggestions for Authors

The authors did great jobs on summarizing the tissue engineering materials on penile/urethral surgeries, and I agree to publish this review. 

Reviewer 3 Report

Comments and Suggestions for Authors

Elia et al in manuscript entitled Tissue Engineering for Penile Reconstruction raise a very interesting topic. Reconstructive techniques in the treatment of penile tissues are constantly evolving and the number of patients who can undergo surgical treatment, e.g. patients with penile cancer, is increasing.

First of all, I would like to commend the authors for the enormous amount of work they put into writing this manuscript - unfortunately, sometimes completely unnecessary.

The manuscript is definitely too long - chapters 3 and 4 are definitely too long, as is chapter 5. All of them should be shortened and, if possible, presented in tabular form. This is not a monograph - the topic of the article is Tissue Enggineering, which is preceded by a huge amount of completely unnecessary data. Before the reader reaches the message and the correct information, he will give up.

Regarding the content in the appropriate chapters, i.e. 6 and 7 - prepared reliably and properly.

To sum up - interesting work, but requires significant changes in content - shortening and removing unnecessary information that does not add significant value in relation to the main topic of the review.

Comments on the Quality of English Language

No comments

Reviewer 4 Report

Comments and Suggestions for Authors

This manuscript aims to review the current status of tissue engineering for penile reconstruction; however it lacks focus partly because there is also a focus on isolated urethral reconstruction as well as the use of biomaterials as part of the management of Peyronies disease.

Hence the information that is summarized into Table 3 seems limited and also lacks focus. Plus there is no data provided on what is the actual need for use of bioengineered tissue in the setting of urethral stricture/Peyronies disease. Because it is often a clinical need that helps act as an ongoing driver to developing solutions to address it

When it comes to the issue of major penile reconstruction, there has been a major review article on what is the current status of the research into bioengineered tissues for this indication which has been published in recent years (which is far more comprehensive)- https://www.nature.com/articles/s41585-019-0246-7

Similarly when it comes to the current results, knowledge pertaining to penile transplantation there are more up to date references. With only 5 cases reported to date this will remain a very uncommon procedure (and will only be able to be delivered in regions where there are the resources as well as the infrastructure in place to facilitate it).

Round 2

Reviewer 3 Report

Comments and Suggestions for Authors

The authors responded correctly to the comments.

The manuscript has improved in quality.

Author Response

We want to thank the reviewer 3 for his/her comments which help us to improve our manuscript.

Reviewer 4 Report

Comments and Suggestions for Authors

The authors have undertaken a number of revisions of the manuscript in line with the comments made by the reviewer.

There remains one issue (which is new). The first sentence in the legend to Table 3 does not quite read correctly. Are the authors trying to imply that the information in Table 3 is not a comprehensive summary of the last decade? If that is the case why is Table 3 an important part of this manuscript?

Round 3

Reviewer 4 Report

Comments and Suggestions for Authors

Can you please ensure that you have uploaded the latest version of the manuscript, which includes all of the alterations mentioned in your cover letter (including the additional references).

Author Response

We upload the latest version with tracks change, please check

Round 4

Reviewer 4 Report

Comments and Suggestions for Authors

I have now looked at the PDF version and am satisfied, that it can be accepted for publication